# Maize/Peanut Intercropping Reduces Carbon Footprint Size and Improves Net Ecosystem Economic Benefits in the Huang-Huai-Hai Region: A Four-Year Study

Zhenhui Yan [1,2], Jianguo Wang [2], Ying Liu [1,2], Zhaoyang You [2], Jialei Zhang [2], Feng Guo [2], Huaxin Gao [2], Lin Li [1,*] and Shubo Wan [1,2,*]

1 College of Agriculture, Hunan Agricultural University, Changsha 410128, China; jluyanzhenhui@163.com (Z.Y.); 18763829271@163.com (Y.L.)
2 Shandong Academy of Agricultural Sciences, Jinan 250100, China; wang_jianguo2020@163.com (J.W.); ricardozyyou@163.com (Z.Y.); zhangjialei19@163.com (J.Z.)
* Correspondence: lilindw@163.com (L.L.); wanshubo2016@163.com (S.W.); Tel.: +86-0531-6665-8127 (S.W.)

**Abstract:** The dual challenges of global climate change and reductions in the amount of arable land represent growing threats to the stability of global human populations. Efforts to further optimize cropping systems to maximize yields while minimizing greenhouse gas emissions in limited land areas have thus emerged as a focus in modern agriculture. Cereal-intercropping management strategies may represent a promising approach to simultaneously addressing both of these challenges in China. We aimed at comprehensively assessing changes in yield, carbon footprint, and net ecosystem economic benefit when transitioning from maize/peanut monoculture to intercropping in a field-scale study in an effort to aid in the development of low-carbon intercropping systems that do not have an adverse impact on Chinese grain yields. Beginning in June of 2018, a randomized complete block design with three treatments was used to initiate this study: (1) peanut monoculture (P), (2) maize monoculture (M), and (3) maize/peanut intercropping (MP). We compared yield, greenhouse gas emissions, carbon footprint and net ecosystem economic benefit. Results over four years showed that the land equivalent ratio associated with MP was greater than 1. All three of these cropping systems were net $CO_2$ and $N_2O$ sources as well as net $CH_4$ sinks, with MP generating significantly ($p < 0.05$) lower $N_2O$ and $CO_2$ flux as well as smaller seasonal $N_2O$ and $CO_2$ emissions relative to M. MP additionally reduced the carbon footprint associated with this cropping system by 11.11–31.65% and 30.37–43.62% relative to M and P, respectively. Consistently, MP treatment resulted in respective 70.69% and 26.25% net ecosystem economic benefit (NEEB) increases relative to the M and P conditions while simultaneously enhancing energy use efficiency. In summary, MP systems have potential economic benefit with lower environmental risk alternative to traditional peanut or maize monocropping systems. Converting from peanut or maize monocropping systems to MP systems practices contributed to improved farmland use efficiency, clean production and increased farmers' income in an agricultural system.

**Keywords:** maize/peanut intercropping; monoculture; crop yield; carbon footprint; net ecosystem economic benefit

## 1. Introduction

Rising levels of anthropogenic greenhouse gas (GHG) emissions including carbon dioxide ($CO_2$), nitrous oxide ($N_2O$), and methane ($CH_4$) are causing progressive damage to natural ecosystems and are expected to have increasingly severe economic and social impacts throughout the globe in the coming years [1]. This process of global climate change represents a threat to humanity that must be addressed [2]. The Paris Agreement established a target goal of ensuring that global temperatures rise by less than 2 °C relative to pre-industrial levels, with specific efforts being made to maintain such warming under

1.5 °C to mitigate the adverse effects of climate change [3]. Agriculture is a major and growing source of human-derived GHG emissions [4,5], with multiple studies having reported a >45% increase in agricultural $N_2O$ emissions since the 1980s [2]. Agriculture-derived GHG emissions have risen by an estimated 10.1% over the last 10 years [6], and now comprise approximately 22% of global emission levels such that they represent a particularly important target for efforts aimed at decreasing GHG production [7,8]. However, the world population is projected to increase further to 9.7 billion in 2050 and 10.4 billion by 2100 [9], while the amount of arable land per capita continues to decline from 0.36 ha per capita in 1961 to just 0.18 ha per capita worldwide and 0.09 ha per capita in China as of 2018 [10]. Efforts to further optimize cropping systems to maximize yields while minimizing GHG emissions in limited land areas have thus emerged as a focus in modern agriculture.

China produced the second-largest amount of maize in the world, with an estimated maize output in 2021 of 261 million tons, comprising 22% of the global maize yield [11]. The Huang-Haui-Hai region of China exhibits ~13 million ha of maize planting area, accounting for roughly 31.66% of the national maize planting area [11], making this region a key source of Chinese grain productivity. China is also the largest global producer of peanuts, having produced 38% of the global peanut supply in 2021 with 50.86% of Chinese peanut planting area being located in the Huang-Huai-Hai region [11,12]. The No.1 central document for 2022 emphasizes the importance of stabilizing grain outputs and sowing area while improving vegetable oil production, highlighting the coordinated development of these strategies as a major national priority important for ensuring Chinese grain and oil security [13].

Intercropping systems have historically been implemented to improve crop yields, and cereal/legume intercropping is firmly established as a sustainable agricultural system that can enhance the efficiency of nitrogen utilization, sunlight utilization, and plant root interactions [14–16]. GHG emissions have also been studied in the context of intercropping systems, revealing that maize/soybean intercropping may allow for reductions in agricultural $N_2O$ output in fertilized planting systems [17]. Legume intercropping or mixed cropping with wheat has similarly been reported to decrease fertilizer-derived $N_2O$ flux [18], while intercropping prairie cordgrass and kura clover can mitigate the $N_2O$ emissions and net warming potential associated with these agrosystems [19]. These results are not universal, however, with some studies instead reporting higher $N_2O$ emissions and yields from intercropping systems. Leguminous intercropping, for example, was found to result in the generation of higher levels of $N_2O$ emissions when developing large biomass levels during dry years in the Great Rift Valley [20]. Alternatively, one study reported that maize/soybean intercropping was associated with a downward trend in the total soil $N_2O$ emission levels relative to monoculture systems, although this difference failed to achieve significance [21]. The relative environmental advantages of intercropping strategies thus warrant further research. Notably, while several studies have examined soybean intercropping strategies, little research focused on the effects of peanut intercropping on crop yields and GHG emissions has yet been performed.

There are two primary approaches to decreasing global GHG accumulation: reducing GHG emissions and enhancing the ability of carbon sinks to absorb these gases, particularly $CO_2$ [8]. Several agronomic parameters impact GHG output in farming settings including irrigation, tillage, and fertilization strategies, as they alter the physicochemical characteristics of the underlying soil. Chemical fertilizers, fossil fuels, machinery, and pesticides used in the context of farming also indirectly contribute to $CO_2$ emissions [22]. The carbon footprint (CF) associated with agriculture has thus been employed as a metric to assess GHG emissions and the environmental benefits of specific agrosystems, quantifying total GHG output in $CO_2$ equivalents ($CO_2$-eq). CF has frequently been used as an index to assess GHG output from different planting systems over time that is used in life cycle assessment (LCA) research focused on low-carbon agriculture [22,23]. CF has also been employed to quantify GHG output from intercropping systems including maize/soybean intercropping in the North China Plain [17], perennial grass/forage legume intercropping [24],

oat/sunflower intercropping [25], and integrated farming with intercropping [26]. These analyses have provided a basis for global efforts to design agricultural systems with a smaller CF. However, no studies have yet been performed using the CF method to assess the environmental impact of maize/peanut intercropping systems in China.

There is often a relationship between GHG emissions and economic benefit such that higher levels of agriculture-related economic gains are likely to incur a greater risk to the environment, in turn decreasing the net economic benefits of such agricultural output [27]. Net ecosystem economic benefit (NEEB) is an index that takes this into account, allowing for the comprehensive assessment of the net economic benefits for a given farming system based on parameters including crop yields, agricultural measures, and CF [28], thereby allowing researchers to effectively gauge whether a given system is sustainable. A field study is thus warranted to assess the sustainability of maize/peanut intercropping as a strategy that can lower GHG emissions and CF size while improving the associated NEEB.

Accordingly, this study entailed a field study of a maize/peanut intercropping system in the Huang-Huai-Hai region of China over a four-year period. The goals of this study were to (1) compare relative maize and peanut yields, (2) assess GHG emissions, (3) examine changes in CF, and (4) assess the NEEB under these different monocropping or intercropping systems. This study was thus aimed at comprehensively assessing changes in CF and NEEB when transitioning from maize/peanut monoculture to intercropping in a field-scale study in an effort to aid in the development of low-carbon intercropping systems that do not have an adverse impact on Chinese grain yields.

## 2. Materials and Methods

### 2.1. Experimental Site

This field study was performed from June 2018–October 2021 at the Jiyang Experimental Base of Shandong Academy of Agricultural Sciences (116°58′ E, 36°58′ N, Figure 1). This study site is 56 m above sea level and subject to a warm temperate continental monsoon climate, with average annual sunshine hours, temperatures, and precipitation levels of 2616.6 h, 12.8 °C, and 580 mm, respectively. Meteorological data for the maize and peanut growing seasons during the study period from the Shandong Meteorological Bureau are shown in Figure 2. The study site soil was brown loam, with a surface layer (0–40 cm) containing respective total nitrogen, organic matter, alkali-hydrolyzable nitrogen, available phosphorus, and available potassium levels of 0.81 mg kg$^{-1}$, 10.23 g kg$^{-1}$, 50.07 mg kg$^{-1}$, 44.70 mg kg$^{-1}$, and 220.13 mg kg$^{-1}$, respectively. Local cropping strategies in the Huang-Huai-Hai region primarily consist of maize/wheat rotations, with wheat being planted in mid-October and harvested in June, while maize is planted following wheat straw return to the field and harvested in mid-October. This study was performed following traditional crop sequences such that it was performed from June–October during each of the study years.

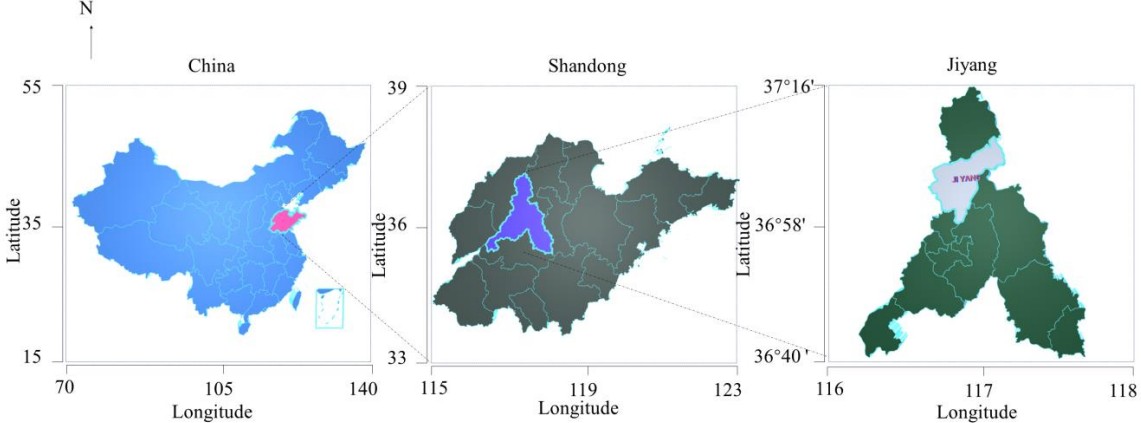

**Figure 1.** Study site location.

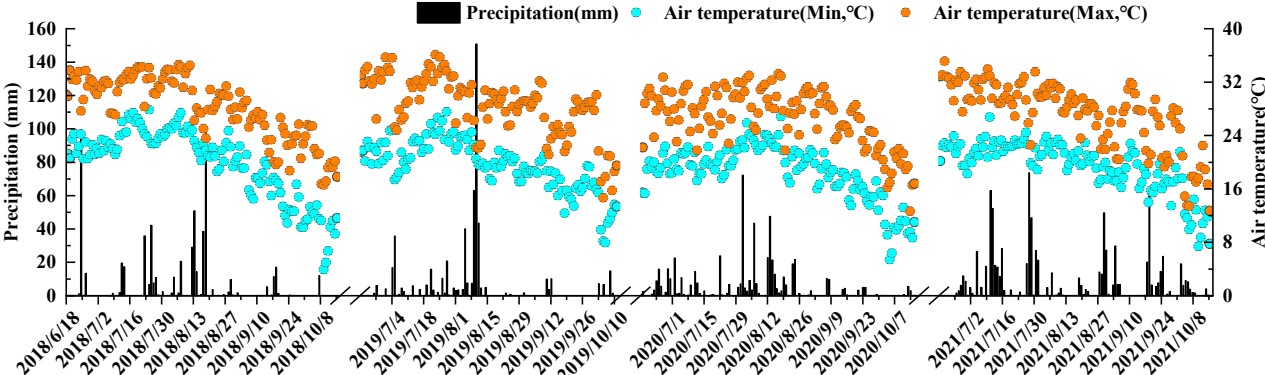

**Figure 2.** Daily precipitation and average daily temperatures during the summer maize and peanut growing seasons.

## 2.2. Experimental Design

Beginning in June of 2018, randomized block design with three replications was used to initiate this study: (1) peanut (*Arachis hypogaea* L. Huayu 25) monoculture (P), (2) maize (*Zea mays* L. Denghai 605) monoculture (M), and (3) maize/peanut intercropping (MP).

The total planting area for this study was 51 m × 45 m, the area of each experimental plot was 17 m × 15 m, planting in the north and south to maximize light energy utilization. Peanuts were planted via ridge sowing (ridge width: 85 cm), with one ridge consisting of two rows, a plant spacing of 10 cm, ridge row spacing of 30 cm, and single-grain precision sowing. For monocropping, maize rows were spaced 55 cm apart with a plant spacing of 26 cm. For MP intercropping, three rows of maize were alternately planted with four rows of peanuts, with a distance of 30 cm between adjacent peanut and maize rows (Figure 3). To ensure that M and MP per unit area plant number is the same, the plant spacing was 13 cm. Prior to sowing, compound fertilizer (750 kg·hm$^{-2}$, N + P$_2$O$_5$ + K$_2$O, N-P-K: 15-15-15) was evenly applied to the soil with rotary tillage. Subsequently, we would like to emphasize that maize plants in monocropping and intercropping systems, nitrogen fertilizer (115 kg·hm$^{-2}$, broadcast fertilization) was top-dressed in the form of urea at the trumpet mouth stage of growth following rainfall > 10 mm, whereas no top dressing was applied for peanut plants. No irrigation was employed, as natural rainfall was sufficient to meet crop needs. Local farming practices were used to guide weeding, pesticide/herbicide application, and other agricultural management strategies (mechanized harvesting). Sowing, topdressing, and harvesting times are compiled in Table A1. During the following year maize and peanut monoculture systems were rotated, while maize and peanut planting areas within intercropping systems were also rotated such that maize planting area from the previous year was converted to peanut planting area and vice versa.

## 2.3. Sampling and Measurements

### 2.3.1. Crop Productivity

When maize and peanut crops were ripe, samples were collected to assess crop yields and aboveground biomass levels. For each monoculture system, 2.2 m$^2$ of maize (2 m long × 2 rows) and 3.4 m$^2$ of peanuts (2 m long × 2 rows) were harvested. For intercropping systems, a 2 m segment was harvested for the maize and peanut strips, each with an area of 3.4 m$^2$. Maize was allowed to air-dry and converted into a standard moisture content (14%) to assess grain yields. Peanuts were allowed to naturally air-dry to a standard water content (10%), after which pod yields were measured.

To assess dry matter and plant numbers, an additional harvest of maize from 0.6 m$^2$ (0.5 m × 1.2 m) and peanuts from 0.8 m$^2$ (0.5 m × 1.6 m) was performed for each plot. These plant samples were heated for 30 min at 105 °C prior to drying to a constant mass for 48 h at 80 °C [29].

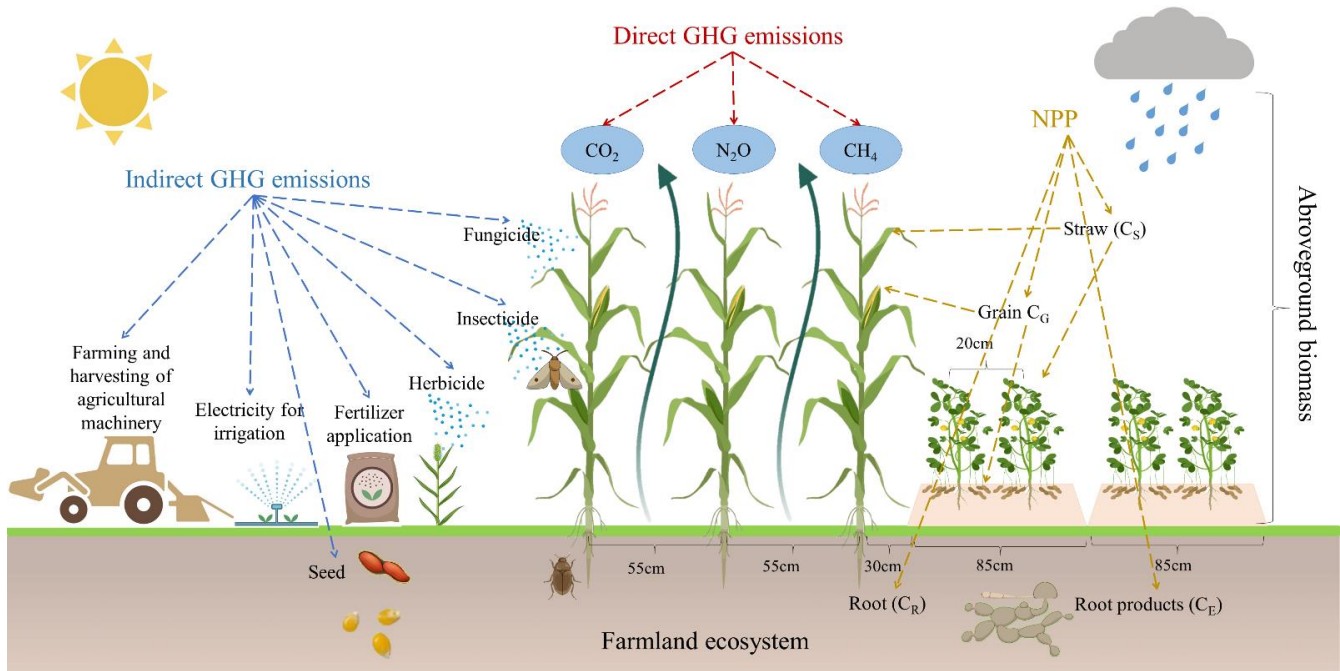

**Figure 3.** Schematic overview of row placement for maize and peanut plants in intercropping systems, carbon sinks and carbon sources taken into consideration when calculating net GHG emissions for a farmland ecosystem. Note: $C_G$, $C_S$, $C_R$, and $C_E$ correspond to harvested product biomass, aboveground residues (primarily straw, litter, and crop residues), roots, and root products (root exudates and fine root turnover), respectively.

The yields of the different cropping systems were converted into maize equivalent yield (MEY) values to facilitate cropping system-level comparisons [30], with maize serving as the reference crop given that it is the most widely cultivated crop in the Huang-Huai-Hai region. MEY is calculated based on crop prices and yields as follows:

$$MEY = P_{peanut}/P_{maize} \times Y_{peanut} \tag{1}$$

where $P_{peanut}$ is the price of peanut seed kernels, $P_{maize}$ is the price of maize, and $Y_{peanut}$ is the yield of peanut seed kernels. Given that both maize and peanuts are harvested every October, the price was calculated during this month.

Land equivalent ratio (LER) values were used to assess the advantages of intercropped land as follows [31]:

$$LER = Y_{im}/Y_m + Y_{ip}/Y_p \tag{2}$$

where $Y_{im}$ and $Y_m$ correspond to the grain yield for the maize belt in the intercropping system and the maize monocropping system, respectively, while $Y_{ip}$ and $Y_p$ similarly correspond to the peanut yields for the intercropping system and the peanut monocropping system, respectively. An LER > 1 indicates that the intercropping system is advantageous, whereas this is not the case when LER < 1.

In order to better explain which crops lead to the increase or decrease in LER, we analyzed the Relative Yield (RY):

$$RY_m = Y_{im}/Y_m \tag{3}$$

$$RY_p = Y_{ip}/Y_p \tag{4}$$

If RY < 0.5, the crop has no intercropping advantage, whereas if RY > 0, the crop is significantly advantaged.

Intercropping advantage (IA, in $kg \cdot hm^{-2}$) is measured as follows:

$$IA = Y_i - (Y_m \times F_m + Y_p \times F_p), Yi = Y_{im} + Y_{ip} \tag{5}$$

where $Y_i$ corresponds to the intercropping system yield, $F_m$ and $F_p$, respectively, denote the proportions of maize and peanut in the intercropping system, such that $F_m = M/(M + P)$, $F_p = P/(M + P)$, where M is the ratio of maize density in the intercropping system to that in the monocropping system, P is the ratio of peanut density in the intercropping system to its single cropping system density. In this study, the maize $F_m$ and peanut $F_p$ values were both 0.5.

The energy yield (GJ/ha) is calculated as follows [32]:

$$EY = Y_g \times E_g + Y_s \times E_s \tag{6}$$

where $Y_g$ and $Y_s$, respectively, correspond to grain and straw yields, while $E_g$ and $E_s$ are the calorific values for grain and straw. Energy yields for the intercropping system were determined by summing individual energy yield values for the peanut and maize crops therein.

### 2.3.2. Greenhouse Gas Emissions

A portable LICA PS-3020 and LICA PS-9000 soil flux measurement system was used to calculate the $N_2O$, $CH_4$, and $CO_2$ emission fluxes. The base for these instruments, which consisted of a polyvinyl chloride pipe (outer diameter: 20 cm, inner diameter: 18 cm, height: 15 cm) was placed in advance such that 5 cm was below ground and 10 cm was on the ground. Three bases were used to collect $N_2O$, $CH_4$, and $CO_2$ in monocropping systems, while three bases were similarly used in different crop belts for the intercropping system that remained in place from soil preparation until harvest. Measurements were made by placing the breathing chamber on the base and collecting $N_2O$, $CH_4$, and $CO_2$ released from the soil. Measurements were made once per week, with three measurements per base. Values were continuously measured from the next day until stable after topdressing, with all measurements being made between 8:30 and 11:00.

Cumulative $N_2O$, $CH_4$, and $CO_2$ emissions were computed via linear interpolation between successive sampling days with the following formula [33]:

$$\text{Cumulative } N_2O, \ CO_2 \text{ or } CH_4 \text{ emissions} = \sum_i^n (F_i \times D_i) \tag{7}$$

where n is the number of sampling intervals. $F_i$ represents the ratio of $NO_2$, $CH_4$, and $CO_2$ emission flux ($N_2O$ and $CH_4$: $nmol \cdot m^{-2} \cdot s^{-1}$, $CO_2$: $\mu mol \cdot m^{-2} \cdot s^{-1}$) within the $i_{th}$ sampling interval, and $D_i$ stands for the number of days between $F_i$ and $F_{i+1}$.

### 2.3.3. Carbon Footprint Calculations

All agronomic ecosystems simultaneously sequester, produce, and consume sources of carbon, and a given crop in a specific planting system can act as a net carbon source or carbon sink at different points in time. As such, many different factors need to be considered when calculating CF values [34]: (a) the carbon input of organic fertilizers, which was 0 in this study ($C_{Import}$, $kg \ CO_2$-eq $ha^{-1}$), (b) the fixed $CO_2$ equivalent in net primary productivity, ($C_{NPP}$, $kg \ CO_2$-eq $ha^{-1}$), (c) the $CO_2$ equivalent of aboveground biomass harvested or removed from the system, ($C_{Export}$, $kg \ CO_2$-eq $ha^{-1}$), (d) GHG equivalents that are directly released from the soil, ($GHG_{Direct}$). $kg \ CO_2$-eq $ha^{-1}$), (e) agricultural inputs such as mineral fertilizers, seeds, pesticides, fuel, plastic film, and electricity, all of which yield indirect GHG equivalents ($GHG_{Indirect}$, $kg \ CO_2$-eq $ha^{-1}$) (Figure 3). Accordingly, CF was calculated as follows:

$$CF_i = C_{Import,i} + C_{NPP,i} + C_{Export,i} + GHG_{Direct,i} + GHG_{Indirect,i} \tag{8}$$

where i denotes the ith treatment. CF values greater than and less than 0 indicate that a given agrosystem is a net carbon source and a net carbon sink, respectively [8].

Net primary productivity is represented by $C_{NPP}$, which takes into account the ability of crops to use photosynthesis to capture and store solar energy, fix $CO_2$ from the air, and store carbon in the form of biomass through the production of grains, straw, roots, and root exudates. NPP was calculated using the following formulas:

$$NPP = C_G + C_S + C_R + C_E$$
$$C_{NPP} = -(C_G + C_S + C_R + C_E) \times 44/12$$
$$C_G = \text{Grain biomass} \times \text{C content}$$
$$C_S = \text{Straw biomass} \times \text{C content} \qquad (9)$$
$$C_R = \text{Root biomass} \times \text{C content}$$
$$C_E = NPP \times 0.11$$
$$C_{Export} = (C_G + C_S) \times 44/12$$

where $C_G$, $C_S$, $C_R$, and $C_E$ correspond to the main product biomass, aboveground residues (primarily straw, litter, and crop residues), roots, and root products (root exudates and fine root turnover) at harvest time, respectively [35]. The ratio of $CO_2$ to C is 44/12. For maize, the carbon content in grains and in the straw/roots can be assumed as 0.39 and 0.47 kg kg$^{-1}$, respectively, while the carbon content in all parts of peanut plants is 0.38 kg kg$^{-1}$ [36]. The additional C produced through root exudates and root turnover is 0.11 kg kg$^{-1}$, which is related to recoverable roots [37,38]. $C_{Export}$ represents the $CO_2$ equivalent corresponding to the amount of harvested or otherwise removed aboveground biomass(main produce and residues) [34].

Levels of emitted $N_2O$ and $CH_4$ were converted into $CO_2$ emissions based upon a 100-year time scale, with $N_2O$ and $CH_4$, respectively, exhibiting warming potentials per unit mass that were 298- and 34-fold higher than those for $CO_2$, [7], with values being reported in kg·hm$^{-2}$.

$$GHG_{Direct} = W_{(CO_2)} + W_{(N_2O)} \times 298 + W_{(CH_4)} \times 34 \qquad (10)$$

where $W_{(CO_2)}$, $W_{(N_2O)}$, $W_{(CH_4)}$, respectively, corresponds to the total $CO_2$, $N_2O$, and $CH_4$ emissions during the growing season.

$$GHG_{Indirect} = \Sigma I_{n,i} \times C_{n,i} \qquad (11)$$

Given that all pieces of land entail different needs for mineral fertilizer, seed, pesticide, fuel, plastic film, and electricity inputs, this results in differences in indirect GHG emissions. In the above formula, $I_{n,i}$ corresponds to the nth quantity of agricultural inputs for the ith treatment as listed in Table A2. $C_{n,i}$ is a coefficient factor corresponding to the nth input that was determined based on previously published studies (Table 1).

**Table 1.** Indirect GHG emission factors for agriculture material inputs used to calculate CF values in the present study.

| Agricultural Inputs | Factors | Units | References |
|---|---|---|---|
| Maize seed | 3.85 | kg $CO_2$-eq kg$^{-1}$ | [35,39] |
| Peanut seed | 0.92 | kg $CO_2$-eq kg$^{-1}$ | [39] |
| Nitrogenous fertilizer (N) | 4.96 | kg $CO_2$-eq kg$^{-1}$ | [8] |
| Compound fertilizer | 1.77 | kg $CO_2$-eq kg$^{-1}$ | [40] |
| Insecticides | 16.60 | kg $CO_2$-eq kg$^{-1}$ | [34] |
| Herbicides | 10.15 | kg $CO_2$-eq kg$^{-1}$ | [34] |
| Fungicide | 10.57 | kg $CO_2$-eq kg$^{-1}$ | [34] |
| Diesel consumption | 4.10 | kg $CO_2$-eq kg$^{-1}$ | [34] |
| Electricity | 1.23 | kg $CO_2$-eq kg$^{-1}$ | [34] |

2.3.4. NEEB Calculations

NEEB serves as an alternative to CF that can be used to specifically evaluate the net economic benefits of particular agricultural systems, and is calculated as follows [41]:

$$NEEB = Yield\ gains - Input_{cost} - CF_{cost} \tag{12}$$

where yield gains correspond to the economics of maize or peanut grain yields based on the prices of these grains in October, given that this is the month when they are harvested each year; $Input_{cost}$ represents the total costs of seeds, fertilizers, herbicides, pesticides, fungicides, diesel, and electricity; and $CF_{cost}$ represents the product of carbon trading price multiply CF, with a carbon trading price of 232.7 CNY (35.1 $) $t^{-1}$ $CO_2$-eq [28,42].

*2.4. Statistical Analysis*

Two-way analyses of variance (ANOVAs) was carried out considering the years and cropping systems as variable factors. Duncan's tests were used to compare data between treatments with SPSS 26.0 (SPSS Inc., Chicago, IL, USA) with $p = 0.05$ and $p = 0.01$ as significance levels. Origin 2022 (OriginLab Corporation, Northampton, MA, USA) was used for figure generation.

**3. Results**

*3.1. Maize and Peanut Yields under Different Cropping Systems*

Yield, RY LER, and intercropping advantage results from 2018–2021 are shown in Table 2, while corresponding EY and MEY results are shown in Figure 4. Relative to monocropping, MP intercropping was associated with an increase in the yield of maize per unit area by 15.79–51.39% for four consecutive years, while the peanut yield per unit area of peanut was reduced by 4.53–19.48%. As the LER was >1, the intercropping system was advantageous with an economic yield advantage of 1420.05–2046.45 kg ha$^{-1}$. However, RY shows that in this intercropping system it is only over-yielding of maize that gives the LER advantage. RY of peanut < 0.50 shows that peanut was significantly disadvantaged, presumably due to shading from the tall maize crops on either side and nutrient competition. The high relative yield of maize contributed most to the high LER. Relative to the M and P systems, the MP system was associated with respective 18.33–36.46% and 61.88–90.75% increases in EY (Figure 4a), and MEY is also the highest among MP.

**Table 2.** Yields, RY, LER and intercropping advantage levels for different cropping systems.

| Year | Treatment | Yield (kg hm$^{-2}$) Maize | RY Maize | Yield (kg hm$^{-2}$) Peanut | RY Peanut | LER | Intercropping Advantage (kg hm$^{-2}$) |
|---|---|---|---|---|---|---|---|
| | M | 10,051.7 a | | - | | - | - |
| 2018 | M\|\|P | 6673.0 b | 0.66 | 1250.2 b | 0.42 | 1.09 | 1420.05 |
| | P | - | | 2954.6 a | | - | - |
| | M | 11,116.7 a | | - | | - | - |
| 2019 | M\|\|P | 7485.7 b | 0.67 | 1290.3 b | 0.40 | 1.08 | 1615.30 |
| | P | - | | 3204.7 a | | - | - |
| | M | 10,990.9 a | | - | | - | - |
| 2020 | M\|\|P | 8597.2 b | 0.78 | 1273.0 b | 0.48 | 1.24 | 2071.85 |
| | P | - | | 2666.7 a | | - | - |
| | M | 8829.1 a | | - | | - | - |
| 2021 | M\|\|P | 6683.2 b | 0.76 | 1090.1 b | 0.42 | 1.17 | 2046.45 |
| | P | - | | 2624.6 a | | - | - |

Note: Different lowercase letters indicate significant differences in yield in the same season across different cropping systems at $p < 0.05$.

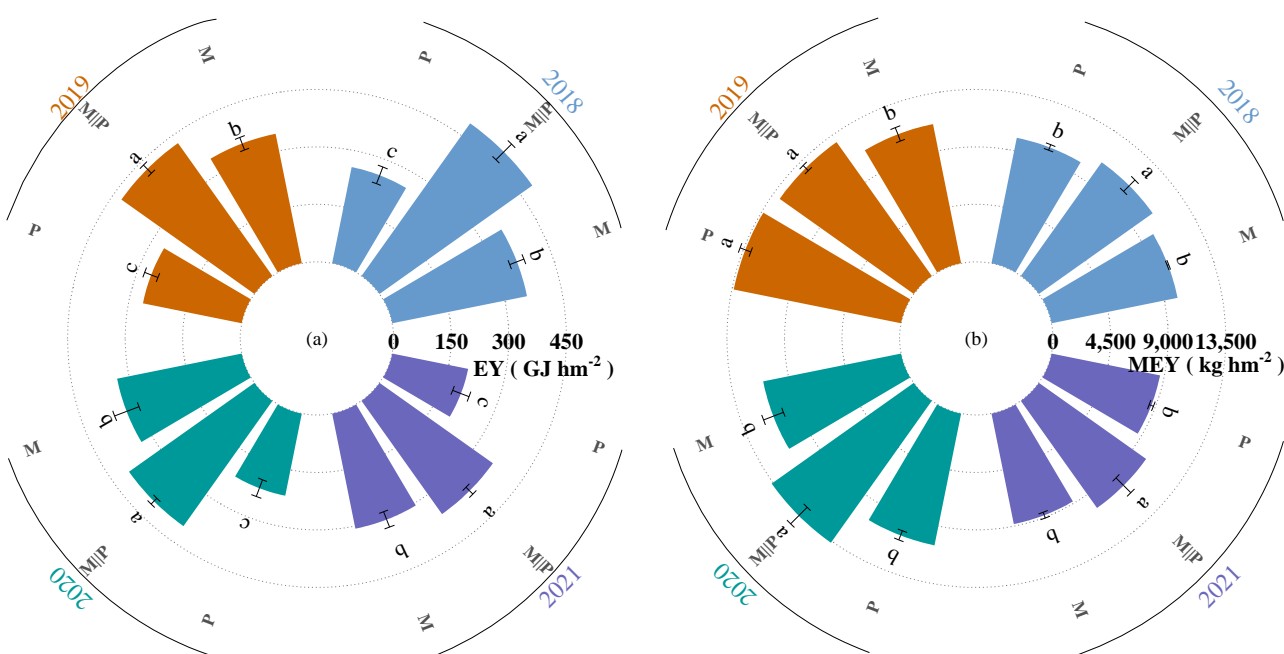

**Figure 4.** Energy yields (EY) (**a**) and maize equivalent yields (MEY) (**b**) for different cropping systems. Note: Different lowercase letters in figure (a, b, c) indicate significant differences in EY and MEY in the same season across different cropping systems at $p < 0.05$.

### 3.2. Direct Soil GHG Emissions

As shown in Figure 5a, $CO_2$ emission flux was monitored for all three planting systems over the four growing seasons, revealing net positive $CO_2$ emission levels under all treatments and during all years consistent with agricultural soil being a net source of $CO_2$. Changes in soil $CO_2$ emission under the three planting systems were largely similar, initially increasing and then decreasing over the growth period, particularly following topdressing. For example, in 2020, the $CO_2$ emission rate after sowing was 2.02–3.59 $\mu mol \cdot m^{-2} \cdot s^{-1}$, and this rate increased slowly with rising temperatures to a peak on day 2 after topdressing, with this peak lasting for roughly 7 days. The peak emission level for the M system was 12.61 $\mu mol \cdot m^{-2} \cdot s^{-1}$, while for the MP system the peak level 9.16 $\mu mol \cdot m^{-2} \cdot s^{-1}$, remaining at roughly 4.50 $\mu mol \cdot m^{-2} \cdot s^{-1}$ after one week and gradually decreasing further to lower levels at maturity. As peanut crops did not require top dressing, emission rates remained relatively stable prior to maturity. With respect to the average emission rates and cumulative emission levels for $CO_2$ (Table 3 and Figure 6a), the M system exhibited the highest values of 17.27 ± 3.06 t ha$^{-1}$ season$^{-1}$ and 4.18 ± 0.04 $\mu mol \cdot m^{-2} \cdot s^{-1}$ averaged over the four-year study period. The next highest values were observed for the MP system, with respective values of 15.58 ± 2.94 t ha$^{-1}$ season$^{-1}$ and 3.77 ± 0.77 $\mu mol \cdot m^{-2} \cdot s^{-1}$ that were 9.78% and 9.80% lower than those values for the M system. Values for the P system were the lowest of the three analyzed cropping systems.

Soil $N_2O$ emission flux (Figure 5b) was monitored under these three cropping systems across four consecutive planting years, similarly only revealing net positive $N_2O$ emission levels consistent with the role of agricultural soil as a $N_2O$ source. Average annual trends were largely similar, with higher $N_2O$ emission flux following base fertilizer application in mid-late June that declined to lower levels after 1–2 weeks. Following maize top dressing in mid-August, an $N_2O$ emission peak was evident for 2–3 days before these emissions gradually declined and were maintained for roughly 1–2 weeks. Emission rates during the off-peak periods remained low under all three cropping systems, demonstrating that agricultural management practices, and particularly nitrogen fertilizer application, had a major impact on soil $N_2O$ emission. The average emission rate and total emission levels for $N_2O$ were highest for the M system, followed by the MP and P systems. The average

emission rate for the P system ($0.27 \pm 0.07$ nmol·m$^{-2}$·s$^{-1}$) was lower than that for the MP system ($0.46 \pm 0.11$), with the M system exhibiting the highest value ($0.64 \pm 0.08$) (Table 3). Similarly, the total emission levels for the P system were 56.98% and 41.23% lower than those for the M and MP systems, with the MP system exhibiting total emission levels 26.79% lower than those for the M system (Figure 6b).

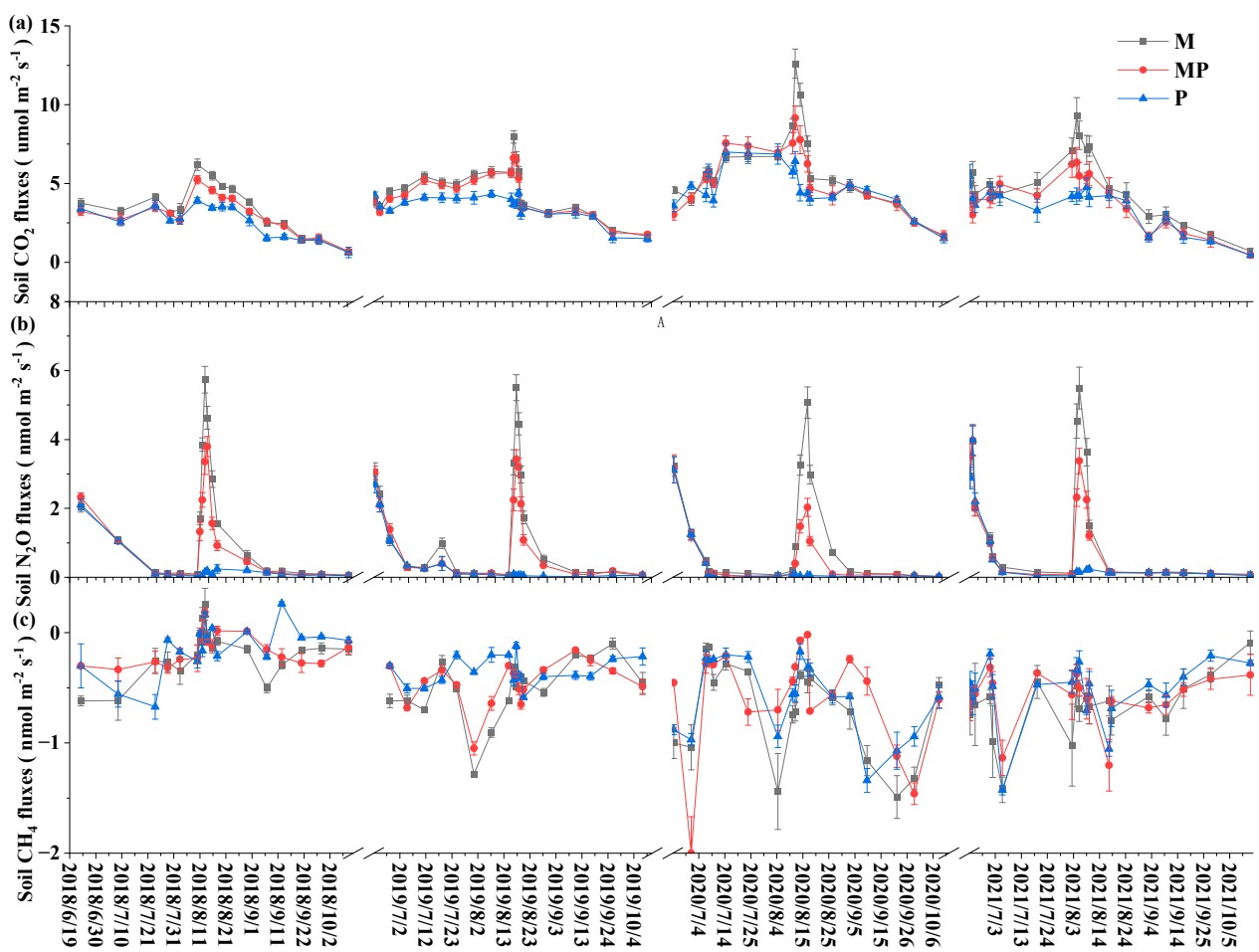

**Figure 5.** Soil $CO_2$ (**a**), $N_2O$ (**b**), and $CH_4$ (**c**) fluxes under different cropping systems from 2018–2021.

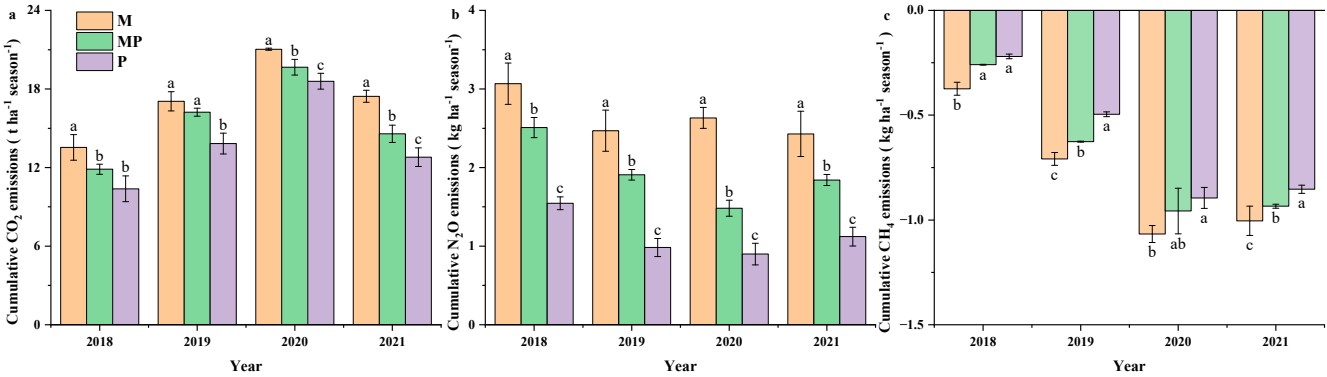

**Figure 6.** Cumulative $CO_2$ (**a**), $N_2O$ (**b**), $CH_4$ (**c**) emissions under different planting systems from 2018–2021. Note: Different lowercase letters in figure (a, b, c) indicate significant differences in Soil $CO_2$ (**a**), $N_2O$ (**b**), and $CH_4$ (**c**) fluxes at $p < 0.05$.

**Table 3.** Averaged soil $CO_2$, $N_2O$, and $CH_4$ emission rates under different cropping systems from 2018–2021.

| Year | Treatments | $CO_2$ Emission Rate (kg hm$^{-1}$ h$^{-1}$) | $N_2O$ Emission Rate (g hm$^{-1}$ h$^{-1}$) | $CH_4$ Emission Rate (g hm$^{-1}$ h$^{-1}$) |
|---|---|---|---|---|
| | M | 5.23 ± 0.30 a | 1.12 ± 0.03 a | −0.14 ± 0.02 b |
| 2018 | MP | 4.58 ± 0.13 b | 0.97 ± 0.03 b | −0.10 ± 0.01 a |
| | P | 4.01 ± 0.16c | 0.60 ± 0.05 c | −0.08 ± 0.01 a |
| | M | 6.46± 0.10 a | 0.93 ± 0.06 a | −0.26 ± 0.01 c |
| 2019 | MP | 6.15 ± 0.14 b | 0.73 ± 0.03 b | −0.24 ± 0.01 b |
| | P | 5.24 ± 0.14 c | 0.38 ± 0.03 c | −0.18 ± 0.01 a |
| | M | 8.05 ± 0.27 a | 1.01 ± 0.03 a | −0.41 ± 0.02 b |
| 2020 | MP | 7.51 ± 0.24 b | 0.57 ± 0.03 b | −0.36 ± 0.04 ab |
| | P | 7.11 ± 0.13 b | 0.35 ± 0.05 c | −0.34 ± 0.02 a |
| | M | 6.73 ± 0.16 a | 0.90± 0.05 a | −0.37 ± 0.02 c |
| 2021 | MP | 5.97 ± 0.32 b | 0.68 ± 0.03 b | −0.35 ± 0.01 b |
| | P | 5.32 ± 0.13 c | 0.41 ± 0.02 c | −0.32 ± 0.01 a |

Note: Different lowercase letters indicate significant differences in averaged soil $CO_2$, $N_2O$, and $CH_4$ emission rates in the same season across different cropping systems at $p < 0.05$.

Both negative and positive $CH_4$ fluxes were detected in all growth seasons (Figure 5c), unlike the measured $N_2O$ and $CO_2$ fluxes. Negative $CH_4$ fluxes were measured on over 90% of sampling days, suggesting that this crop soil is a net $CH_4$ sink. While the differences among systems were not significant during all study years, the overall trend with respect to cumulative $CH_4$ absorption during the four-year study period was as follows: M > MP > P.

*3.3. $CO_2$ Fixation of NPP, Equivalent $CO_2$ Emissions, and Composition*

The fixation of $CO_2$ in NPP for the three cropping systems from 2018–2021 are shown in Figure 7a. Significant differences were observed among the three $C_{NPP}$ values in 2020 and 2021 ($p < 0.05$), with the P system exhibiting the lowest values followed by the M and MP systems. During these two consecutive years, the values for the MP system were 24.64% and 12.29% higher than for the M system and 153.69% and 115.99% higher than for the P system. While the M and MP system values were not significantly different in 2018 and 2019, the $C_{NPP}$ of the MP system during these years was 8.87% and 6.64% higher than the M system, and 146.36% and 143.39% higher than the P system.

$C_{Export}$ exhibited similar trends to $C_{NPP}$ values across these cropping systems (Figure 7a), with significant differences among treatments from 2018–2020. $C_{Export}$ values for the MP system were highest, followed by the M and P systems. While the M and MP systems did not exhibit significant differences in these values in 2018, there remained a significant difference between the M and P systems. Average $C_{Export}$ values during this four-year study for the MP, M, and P systems were 37,322.39 ± 5058.29, 32,086.45 ± 2017.46, 16,864.63 ± 2317.34 kg $CO_2$-eq ha$^{-1}$, respectively.

With respect to soil direct GHG emissions (Figure 7b), these cropping systems are net $CO_2$ and $N_2O$ sources, with total $CO_2$ emission levels of 10,379.17–21,032.47 kg ha$^{-1}$ season$^{-1}$ and $N_2O$ emission levels of 0.90–3.07 kg ha$^{-1}$ season$^{-1}$. $CH_4$ was primarily absorbed in this system, with a net absorption level of 0.06–1.20 kg ha$^{-1}$ season$^{-1}$. Based on the total levels of GHG emissions from these systems, average annual trends were similar across years, with significant differences among systems ($p < 0.05$). The emission levels were lowest for the P system (14,219.53 ± 3362.20), followed by the MP system (16,141.09 ± 3126.49) and the M system (18,037.44 ± 3003.31).

The intercropping system in this study exhibited the highest $GHG_{Indirect}$ (Figure 7c), followed by the P system, with the M system exhibiting the lowest value. The total $GHG_{Indirect}$ is about 3471 kg ha$^{-1}$ season$^{-1}$, with indirect emissions from fertilizer, diesel consumption, and electricity making up 82.82–92.74% of these emissions.

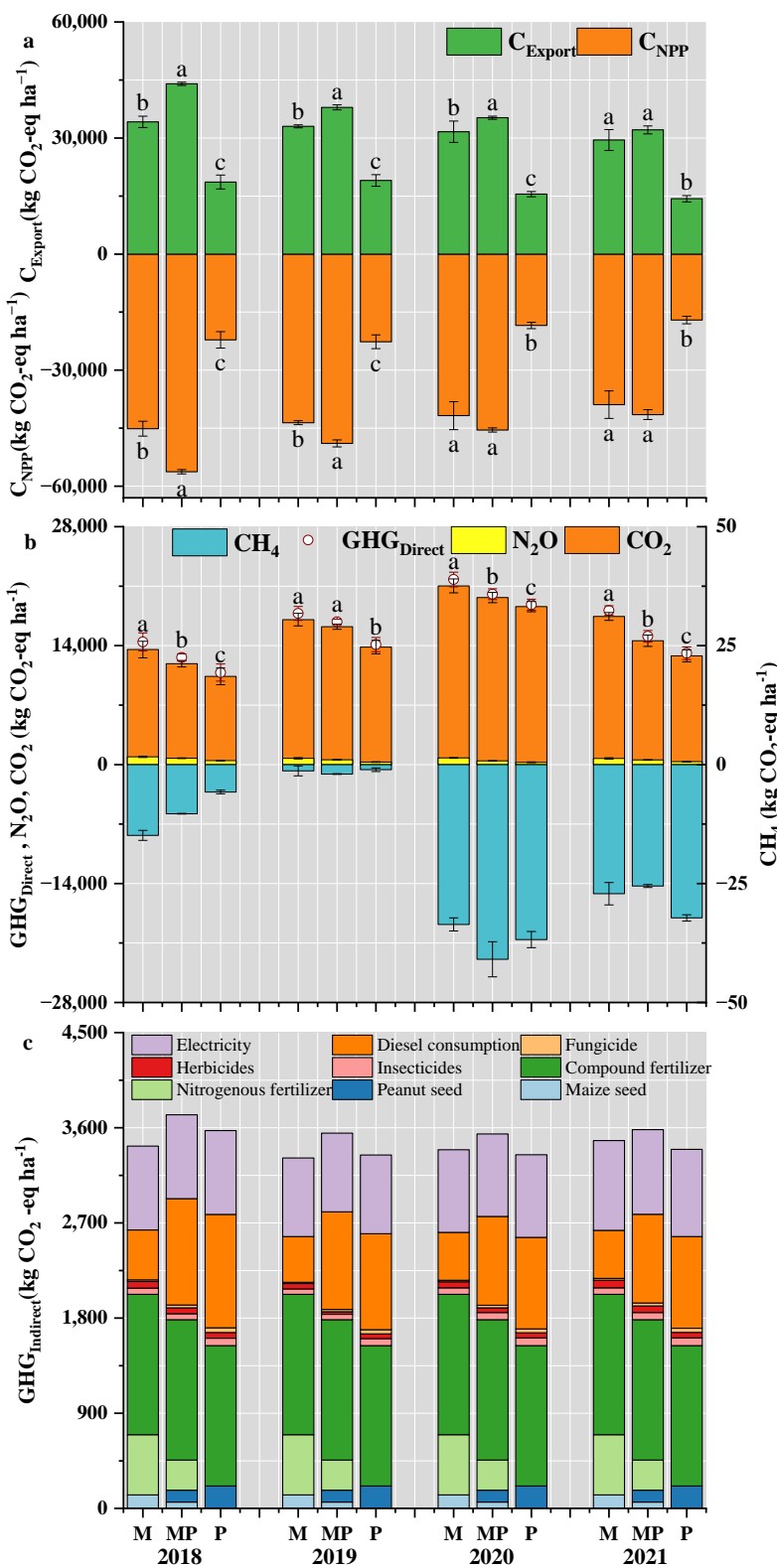

**Figure 7.** Carbon footprint (CF) (kg $CO_2$-eq ha$^{-1}$) calculations were made based on the $CO_2$ fixation of net primary productivity ($C_{NPP}$), (**a**) $CO_2$ equivalents from the harvested products ($C_{Export}$), (**a**) direct GHG emissions (GHG$_{Direct}$), (**b**) and indirect GHG emissions (GHG$_{Indirect}$), (**c**) under different cropping systems from 2018–2021. Note: Different lowercase letters in figure (a, b, c) indicate significant differences in $C_{NPP}$, $C_{Export}$ and GHG$_{Direct}$ in the same season across different cropping systems at $p < 0.05$.

### 3.4. Comparison of the Carbon Footprint Size Associated with Different Cropping Systems

Net GHG emissions were calculated in the form of CF values based on $C_{NPP}$, $C_{Export}$, $GHG_{Direct}$, and $GHG_{Indirect}$ values (Figure 8a). The CF values for the three cropping systems ranged from 3608.66 kg ha$^{-1}$–19,500.71 kg ha$^{-1}$ at the time of harvesting. Over the four-year study period, the P system exhibited the highest CF (14,405.59 $\pm$ 3209.36 kg ha$^{-1}$), followed by a 22.42% lower CF for the M system (11,175.47 $\pm$ 3142.41 kg ha$^{-1}$), while the MP system exhibited the lowest CF (9025.38 $\pm$ 3460.88 kg ha$^{-1}$), with this value being 37.35% and 19.24% lower than the respective CF values for the P and M systems.

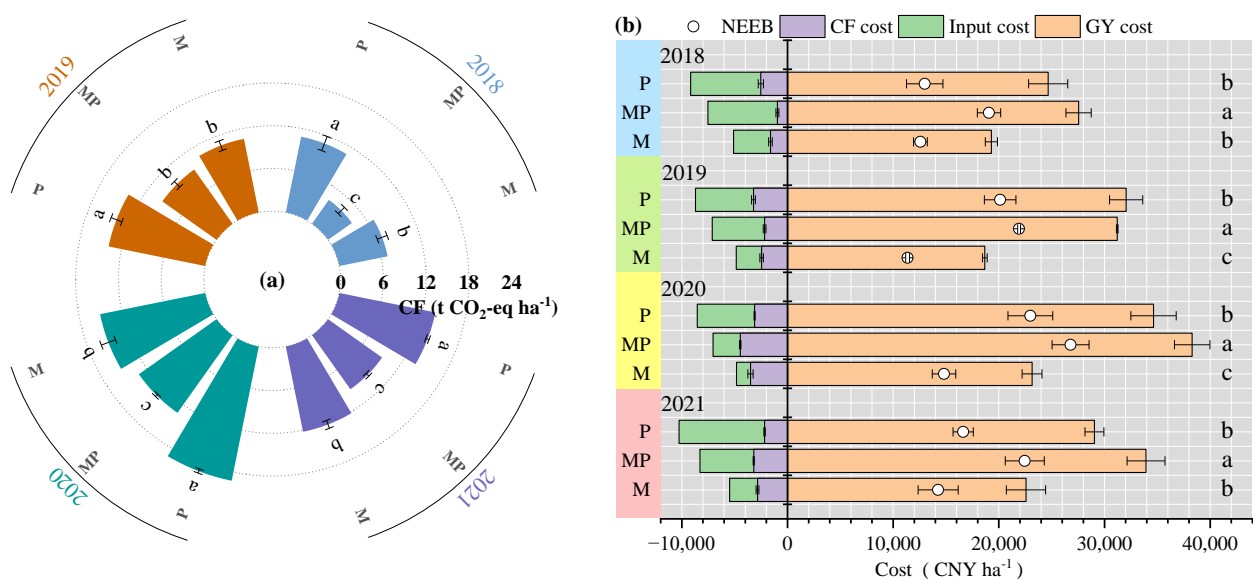

**Figure 8.** CF (**a**) and NEEB (**b**) during crop growing seasons from 2018–2021 under different cropping systems. Note: Different lowercase letters in figure (a, b, c) indicate significant differences in CF, and NEEB in the same season across different cropping systems at $p < 0.05$.

### 3.5. Net Ecosystem Economic Benefit Calculations

Lastly, the NEEB at the time of crop harvest was calculated for each of these cropping systems based on grain yields, agricultural input costs, and CF (Figure 8b). NEEB values varied across cropping systems and seasons, ranging from 11,365–26,789 CNY ha$^{-1}$. The P system exhibited the highest agricultural input costs (8541–10,265 CNY ha$^{-1}$), although the fact that peanuts exhibit a unit price 3.2- to 4.2-fold higher than that of maize resulted in no significant difference in the NEEB of the M and P systems in 2018 or 2021, although these values did differ significantly in 2019 and 2020. The NEEB of MP was significantly higher than that for the M or P systems owing to higher yields. Together, these results demonstrate that this MP intercropping system can enhance the economic benefits to farmers without incurring higher environmental costs.

## 4. Discussion

### 4.1. The Relationship between Intercropping and Crop Productivity

The symbiotic nitrogen-fixing characteristics of leguminous crops have led to their common use in intercropping systems throughout China, Africa, the Americas, and Southeast Asia. Relative to monocropping systems, intercropping with legumes can stabilize yields or improve overall yield levels while enhancing land, solar energy, and resource utilization efficiency, resulting in a net increase in farmer income. Intercropping is thus regarded as a sustainable agronomic practice. Recent studies have demonstrated the yield advantages associated with intercropping systems including maize/soybeans, maize/rapeseed, sugarcane/soybeans, and maize/wheat in China, sunflower/legumes in Serbia [43], sugarcane/green onions [44], and sesame/millet [45] in India. Consistently, in this study the

MP intercropping system exhibited a LER greater than 1 during four consecutive years, highlighting the yield advantages associated with maize/peanut intercropping. The yield of intercropped soybeans is about 1590 kg $hm^{-2}$, while the yield of intercropped peanuts is about 1225 kg $hm^{-2}$. It is 29.8% higher than peanut. Using 2020 as an example, the price of peanut was 9406.7 CNY $t^{-1}$, while the price of soybean was only 5047.5 CNY $t^{-1}$. The economic benefit of peanut in intercropping system was 43.6% higher than that of soybean. These results thus demonstrate that the MP system can enhance crop productivity relative to the M and P monocropping systems.

To more fully assess the yield advantages associated with the MP system, energy yield levels were assessed by multiplying grain and straw yields by the corresponding calorific values. Here, the MP system exhibited significant improvements in energy yield levels relative to the M and P monocropping systems. As a C4 crop, maize exhibits a higher rate of photosynthesis than C3 (generally) plants, exhibiting advantages with respect to the conversion of solar energy to plant quantity [46]. Peanuts exhibit lower energy yields, primarily owing to lower biological peanut plant yields. Maize monoculture has previously been shown to be associated with higher energy yields than maize/peanut intercropping systems [32], in contrast to the present results. This difference may be attributable to the fact that the number of plants per unit area of maize in this experimental intercropping system was consistent with that in the monoculture systems, whereas in their prior experiment the authors maintained the same plant and row spacing, thus resulting in a net decrease in maize per unit area in their intercropping system. Maize energy yields in the intercropping system were lower than those for the monocropping system although the opposite was true for peanut energy yields, resulting in a significantly higher net energy yield for the MP system. Similarly, maize/wheat, maize/pea, and maize/rape intercropping systems have been reported to exhibit higher energy yields than corresponding monocropping systems [47].

### 4.2. GHG Emissions from Maize and Peanut Intercropping System

Different cropping systems require the implementation of different land management strategies that can significantly impact soil biogeochemical processes, crop biology, and system outputs [48]. Key factors that affect GHG source/sink intensity in agricultural farmland include land use patterns, climatic factors, agricultural production level and structure, soil conditions, and crop growth. Scientific debate remains regarding whether intercropped pulse crops can decrease GHG emissions owing to the publication of conflicting results [40,49]. Shen et al. (2018), for example, observed lower $N_2O$ flux and seasonal $N_2O$ emission levels for a maize/soybean intercropping system as compared to a maize monocropping system in the North China Plain [17]. The intercropping of prairie cordgrass and kura clover was further shown by Abagandura et al. (2020) to result in improved biomass yield together with simultaneous reductions in fertilizer-derived $N_2O$ emissions and net global warming potential [19]. Raji et al. (2020) found that maize/*C. juncea* or *L. purpureus* intercropping was still associated with the risk of elevated levels of $N_2O$ emissions, particularly during drier years [20]. Canisares et al. (2021) further demonstrated that the intercropping of maize and *B. humidicola* led to increases in the emission of $N_2O$ as compared to maize monocropping [50]. As such, a range of factors can influence the findings of these different intercropping studies including climatic conditions, crop types, planting density, agronomic management measures, and the testing approaches and intervals. Given that agricultural systems require extended periods of time, only long-term soil GHG emission monitoring for over 10 years can provide accurate insight into the true benefits or limitations of intercropping [51].

The present results confirmed the hypothesis that maize/peanut intercropping would result in reductions in GHG emissions as compared to the monocropping of either species in isolation. Soil $CO_2$ emission results from a series of complex biological and biochemical processes, with soil microbial respiration and the oxidative decomposition of soil organic matter as the two major sources of $CO_2$ release. The study site entered a rainy period

beginning from late July to early August, with concomitant increases in temperature driving enhanced soil microbe activity, organic matter decomposition, and respiration activity for both crop roots and soil microbes. During early August, topdressing provides high levels of nutrients that can support microbial activity and root growth, with root exudates in turn enhancing microbial respiration such that net soil respiration rises to peak levels in mid-August. After this period, crop growth slows and root activity decreases, while temperatures, rainfall levels, and soil water content gradually decrease. Most studies have suggested that the intercropping of grasses and legumes primarily impacts total soil $CO_2$ emissions through changes in nitrogen fertilizer input and soil organic carbon (SOC) fixation and mineralization. Relative to Gramineae monocropping, Gramineae-legume intercropping can lower the need for nitrogen fertilizer application and decrease energy consumption without adversely impacting crop yields, thus reducing $CO_2$ emissions. In addition, the reasonable intercropping of grasses and legumes can enhance crop resource utilization efficiency through improvements in the land multiple cropping index, producing a higher crop biomass than that achieved through monocropping. Straw return can also enhance organic matter input in the soil, raise SOC levels, and decrease $CO_2$ emissions.

Total $N_2O$ emissions during the growing season in the present study ranged from 0.90–3.07 kg ha$^{-1}$ season$^{-1}$ (Figure 8), with lower $N_2O$ emission fluxes from the MP system relative to the M system, thus decreasing total seasonal $N_2O$ emissions. This is largely driven by the fact that the intercropping system employs a topdressing amount half that of the monocropping maize system, as many studies have demonstrated a positive correlation between nitrogen fertilizer application rates and $N_2O$ emissions [52]. In contrast, peanut monocropping only required basal fertilizer application without any topdressing such that nitrogen fertilizer levels were lower than those for the M or MP systems. In the intercropping system, peanuts were cultivated in the soil in place of maize, and peanuts can fix nitrogen from the air, generating significantly lower levels of $N_2O$ than nitrogen fertilization. Given that these two crops can use nitrogen in a complementary manner, the need for exogenous fertilizer was decreased such that intercropping can minimize nitrogen application while maintaining crop productivity. While the $N_2O$ emission levels were lower than those for $CO_2$, the warming potential of $N_2O$ is 298-fold higher than that of $CO_2$ and it cannot be decomposed in the air, contributing to ozone layer destruction.

In this study, the analyzed cropping systems served as a net $CH_4$ sink. As methanogenic bacteria can only release $CH_4$ under anaerobic conditions, proper soil aeration can interfere with the activity of these microbes while benefiting methane-oxidizing bacteria, thereby enhancing net $CH_4$ absorption. Rain and heat levels are maximal during the same period in this study region, and rain is the main source of irrigation. As such, GHG emissions during the study period were primarily derived from $CO_2$, with the soil acting as a net $CH_4$ sink.

### 4.3. The Impact of Intercropping on CF and NEEB

While several prior studies have assessed carbon sequestration and GHG emissions under a range of cropping systems [53], few articles have focused on GHG emissions associated with all possible sinks and sources in MP intercropping systems. Here, CF values were calculated based on $C_{NPP}$ (−17.05 to −56.26 t $CO_2$-eq ha$^{-1}$), $C_{Export}$ (15.50 to 44.02 t $CO_2$-eq ha$^{-1}$), $GHG_{Direct}$ (10.83 to 21.78 t $CO_2$-eq ha$^{-1}$), and $GHG_{Indirect}$ (3.31 to 3.72 t $CO_2$-eq ha$^{-1}$), ranging from 4.73 to 19.21 t $CO_2$-eq ha$^{-1}$ (Figure 8a). A large proportion of these CF values were based on $C_{Export}$, which represents the $CO_2$ equivalent corresponding to the aboveground biomass harvested or removed from the system such that higher output will increase the $C_{Export}$ value, with this being desirable in an agricultural production setting. NPP (grain, straw, root, and root exudates) will also be higher for intercropping systems relative to monocropping systems, yielding a higher $C_{NPP}$. Moreover, $GHG_{Direct}$ accounts for a large proportion of total emissions, suggesting that reducing direct GHG emissions will be an effective means of fostering low-carbon agricultural practices in the Huang-Huai-Hai region. In general, the growth of more crops and higher levels of agricultural inputs are associated with increased $GHG_{Indirect}$ emissions [8]. So the

intercropping system had the highest $GHG_{Indirect}$. The $GHG_{Indirect}$, in contrast, accounted for a relatively small proportion of the CF with chemical fertilizers accounting for the largest proportion thereof (Figure 7c), particularly in the M system, accounting for about 55% for four consecutive years, followed by 45% and 39% in the P and MP systems. P system of the $GHG_{Direct}$ is the smallest (Figure 7b), and the value of $C_{NPP}$ and $C_{Export}$ is also small (Figure 7a). However, according to Formula 8, the sum of $C_{NPP}$ and $C_{Export}$ is the largest. Its sum is 6661.5–8681.0 kg $CO_2$-eq $ha^{-1}$ more than M and MP, resulting in the highest CF value of P. These analyses thus indicated that the intercropping of maize and peanuts was sufficient to lower GHG emissions and decrease the associated CF size as compared to the monocropping of either of these plants.

There is a growing focus on the utility of NEEB analyses as a means of gauging the social and economic benefits associated with particular agricultural systems. NEEB values are determined by taking factors including yields, CF, and the costs of agricultural inputs into consideration [28]. Xu et al. assessed NEEB when converting paddy fields from double-season rice to ratoon rice [34]. Many articles have recently explored the relative CF of specific intercropping systems, with a particular focus on grass and soybean intercropping. Wang et al., for example, assessed CF values associated with sugarcane/soybean intercropping systems using different levels of nitrogen fertilizer input [40]. In contrast, there have been few field studies to date assessing NEEB outcomes associated with a switch from monocropping to intercropping systems using peanuts. Here, the MP system was associated with a higher NEEB relative to the P or M systems (Figure 8), suggesting that this MP system should be proposed to farmers as a strategy that can enhance economic efficiency. While the total seed, fertilizer, pesticide, fungicide, herbicide, diesel, and electricity costs for the M system were low, the $CF_{cost}$ was relatively high such that the economic benefits were lower than for the intercropping system. The economic benefits of the MP system were primarily tied to relatively high maize and peanut grain yields together with a lower $CF_{cost}$. As mentioned earlier, 13 million hectares of maize are planted in this region. If all of them are changed to maize/peanut intercropping, every year the CF will be reduced by 279.5 million tons, and an increase in NEEB by 120.9 billion CNY. Owing to the COVID-19 pandemic and constantly shifting domestic and foreign conditions, agricultural material prices varied significantly over the study period. Peanut prices, for example, rose by 51.13% in 2020 relative to 2018, while nitrogen fertilizer prices in 2021 were 54.55% higher than in 2020, contributing to significant year-to-year variations in NEEB results.

## 5. Conclusions

Climate change, rising global food demands, and decreasing amounts of arable land have emerged as increasingly serious threats to societal integrity over recent decades, underscoring the need for the development of a diverse array of practical solutions to these increasingly intractable issues. The optimization of agronomic management practices and cropping systems may represent a promising means of reducing GHG emissions and saving energy, thereby yielding net benefits without compromising crop productivity. In this study, a maize/peanut intercropping system was assessed under field conditions in the Huang-Huai-Hai region in China over a four-year period. In this setting, maize/peanut intercropping significantly improved crop productivity per unit of arable land while requiring lower levels of nitrogen application, suggesting that this or similar maize intercropping systems may help mitigate the growing strain on arable land area. Moreover, relative to the M system, MP was associated with reduced GHG emissions, with a particularly pronounced 27.17% drop in $N_2O$ emissions over four years. MP conditions also decreased the associated CF by 11.11–31.65% and 30.37–43.62% relative to M and P, respectively, with corresponding increases in NEEB of 70.69% and 26.25% and improved energy use efficiency. These findings suggest that converting from peanut or maize monocropping systems to maize/peanut intercropping systems may represent a promising means of shrinking the agronomic CF while increasing NEEB levels in the Huang-Huai-Hai region of China.

**Author Contributions:** Z.Y. (Zhenhui Yan) and J.W.: Experiment design, writing—original draft preparation, sampling and analysis. S.W.: investigation, funding acquisition. Y.L. and F.G.: Writing—reviewing. Z.Y. (Zhaoyang You) and H.G.: Writing—original draft. J.Z.: Data analysis. L.L.: Editing, project administration, supervision, funding acquisition. All authors have read and agreed to the published version of the manuscript.

**Funding:** This research was funded by Natural Science Foundation of Shandong Province (ZR2020MC094), the earmarked fund for CARS-13, the Taishan Scholars Program and Key R&D program of Shandong Province (2021LZGC026-04).

**Data Availability Statement:** Most of the collected data are contained in the tables and figures in the manuscript.

**Conflicts of Interest:** The authors declare that they have no known competing financial interest or personal relationships that could have appeared to influence the work reported in this paper.

## Appendix A

**Table A1.** Time for sowing, dressing and harvesting.

| Year | Sowing Time | Topdressing Time | Harvest Time |
|------|-------------|------------------|--------------|
| 2018 | 23 June | 9 August | 9 October |
| 2019 | 21 June | 15 August | 12 October |
| 2020 | 18 June | 10 August | 15 October |
| 2021 | 19 June | 2 August | 16 October |

**Table A2.** Unit prices of agricultural inputs and outputs.

| Items | Unit Prices | | | | Units | Sources |
|-------|------|------|------|------|-------|---------|
| | **2018** | **2019** | **2020** | **2021** | | |
| Maize seed | 34.4 | 34.4 | 34.4 | 37.5 | CNY kg$^{-1}$ | Market price at the time of purchase |
| Peanut seed | 18.0 | 18.0 | 18.0 | 24.0 | CNY kg$^{-1}$ | Market price at the time of purchase |
| Nitrogenous fertilizer (N) | 2073.9 | 1997.5 | 1730.3 | 2674.2 | CNY t$^{-1}$ | http://yte1.com/ (accessed on 10 March 2023) [54] |
| Compound fertilizer | 2565.0 | 2580.0 | 2390.0 | 2601.6 | CNY t$^{-1}$ | http://yte1.com/ (accessed on 10 March 2023) [54] |
| Insecticides | 100.0 | 100.0 | 100.0 | 100.0 | CNY kg$^{-1}$ | [55] |
| Herbicides | 26,812.1 | 24,000.0 | 21,000.0 | 47,656.3 | CNY t$^{-1}$ | http://yte1.com/ (accessed on 10 March 2023) [54] |
| Fungicide | 90.0 | 90.0 | 90.0 | 90.0 | CNY kg$^{-1}$ | [55] |
| Diesel consumption | 6716.1 | 6369.9 | 5275.0 | 6486.0 | CNY t$^{-1}$ | http://yte1.com/ (accessed on 10 March 2023) [54] |
| Electricity | 0.6 | 0.6 | 0.6 | 0.6 | CNY KWH$^{-1}$ | [34] |
| Maize | 1772.4 | 1809.5 | 2302.0 | 2511.5 | CNY t$^{-1}$ | http://yte1.com/ (accessed on 10 March 2023) [54] |
| Peanut | 6010.0 | 7535.6 | 9083.3 | 8283.3 | CNY t$^{-1}$ | http://yte1.com/ (accessed on 10 March 2023) [54] |
| Carbon-trade price | 232.7 | 232.7 | 232.7 | 232.7 | CNY t$^{-1}$ CO$_2$-eq | [28] |

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
