# Peer review of "Maize/Peanut Intercropping Reduces Carbon Footprint Size and Improves Net Ecosystem Economic Benefits in the Huang-Huai-Hai Region: A Four-Year Study"

_agronomy, doi:10.3390/agronomy13051343_

Round 1

Reviewer 1 Report

On the whole this is a good piece of careful agronomic research which moves the science of intercropping forwards. The analyses and LCA are very interesting. I have a few comments.

L50: The authors should cite a primary source, not another research paper. I am surprised that this rather dated statistic is still being quoted. According to the most recent UN estimates, global population in 2050 will be 9.7 billions, an increase of 21% from the current 8 billions. Is it really likely that food demand will double in 27 years? Indeed, the whole food demand in 2050 could be met by current cropping areas by eliminating food wastage in storage and in-field losses. 

L60: Correct the year '202'. Line 425, too.

L158: Is the nitrogenous fertilizer urea? The authors should state this. 

L186: Should 'plan' be 'plant'?

L296: For carbon trading, cite a primary source not another research paper. 

L303: In Table 2, include Relative Yield, see screen shot below with my insertions. LER only tells part of the story.  RY shows that in this intercropping system it is only over-yielding of maize that gives the LER advantage. RY of <0.50 shows that peanut is significantly disadvantaged, presumably due to shading from the tall maize crops on either side. Any farmers adopting this system would have to accept reduced peanut productivity compared to sole cropping. Given the 3.2 - 4.2 greater unit price for peanut, farmers may be reluctant to accept the yield reduction. 

L414: Why did the P system exhibit the highest CF? This is surprising, given that N fertilizer was not applied to the sole P system. The authors need to explain why this is. 

L425: 'MP' instead of 'MB'

L446-448: This sentence needs to be re-written. As a C4 crop, maize exhibits a higher rate of photosynthesis than C3 (generally), not biomass. Many C4 crops have a lower biomass than peanuts, e.g. tef, finger millet. 'Quality' presumably should be 'quantity'. 

L476: Cite the genus - Brachiaria. 

L581: 'MEEB' not 'NEED'

Screenshot

Reviewer 2 Report

This paper discussed the effects of Maize/Peanut intercropping system on carbon footprint and Net Ecosystem Economic Benefits compared to monoculture systems under field conditions over for-year field experiment. The authors provide an interesting effort to link costs, product yield, and quality to environmental performance.

My recommendation is to accept after minor revision.

Below are more detailed comments to support the revision.

Abstract

Remove: Background, Results, SIGNIFICANCE:

Add the acronym for: “net ecosystem economic benefit” (NEEB)

Add the symbol used for: (1) peanut monoculture (P), (2) maize monoculture (M), and (3) maize/peanut intercropping (MP)

Specify: with MP was greater than 1 (P)

Introduction

Line 60: the global peanut supply in 2021 with 50.86%

Line 62: The Central No.1 document produced in 2022 (add a reference)

Materials and Methods

Line 182: The number of ears per row and grains per ear were analyzed on 20 ears per treatment condition

Line 186: To assess (total) dry matter and plant numbers: Remove total

Line 188: add a reference for the procedure used

Formula (3): for the monocropping systems you reported Ym and Yp without “s”

Line 208: specify Yi=YIM+YIP

Formula (6): define GWPDirec or use GHGDirec

Line 263: CExport= (CG+CS)×44/12. It could be useful highlight that it is the biomass harvested.

Line 264: CG….. correspond to the main product biomass

Line 266-267: For maize, the carbon content in grains and in the straw/roots can be assumed as….

270-271: CExport represents the CO2 equivalent corresponding to the amount of harvested or otherwise removed aboveground biomass (main product and residues) [32].

Lines 294-295: rephrase the sentence “CFcost represents the product of carbon trading price and CF” is not clear.

Line 298: Two-way analysis of variance (ANOVA) was carried out considering the years and cropping systems as variable factors. Duncan’s test was used…..

Results

Line 307: Yield, LER, and intercropping advantage results from 2018-2021 are shown in Table 2

Lines 315-316: put in the Discussion section.

Lines 359-362: put in the Discussion section.

Lines 376-377: rephrase. The fixation of CO2 in NPP for the three cropping systems from 2018-2021 are shown in Figure 7a.

Line 378: maybe you are discussing about 2020-2021

Line 382: it is 2018-2019

Line 385: CExport values for the MP system were highest

Lines 386-387: it is 2020

Line 391: With respect to soil direct GHG emissions (Fig. 7b)

Lines 392-395: put in the Discussion section.

Lines 401-402: put in the Discussion section.

Lines 402-402: put in the Discussion section.

Line 412: The CF values for the three cropping systems

Reviewer 3 Report

Thank you for the opportunity to review your manuscript.  This work builds upon research conducted for other intercropping systems in China.  The use of peanuts as a legume intercrop with Maize is new, but limited in large scale feasibility based on peanut development requirements compared to intercropping with soybeans, which has a greater tolerance for suitable environments.  Based on peanut production in China, this study makes sense for an intercropping option.

Overall, I think this manuscript is well crafted.  I only have minor edits and comments which are listed below:

Introduction Section:

Please provide the number of hectares currently in intercropping production in China.  This will help add value to the paper by showing this is becoming an accepted practice.  The practice is alluded to in lines 96-102, but quantifiable numbers would be advantageous to the reader.

Materials and Methods Section:

Please add details about topdress fertilizer used, application method, amount and frequency

Discussion Section:

It would be value added to see how much the numbers or values of GHG emission reduction, overall yield or intercropping advantage align or deviate when Maize/Peanut intercropping is compared to Maize/Soybean intercropping.  You could also compare to any Maize/Legume intercropping system to see a direct comparison of this intercropping system compared to others.  Based on hectares of each crop grown in China, I think this would be value added discussion.

Line 82: change examing to examined

Line 119: remove extra period (.)

Figure 3: Is it possible to replace the tick image with a Maize pest or generic moth, caterpillar, beetle, etc.

Line 284:  Add space between paragraph and Table 1

Line 287:  Add space between Table 1 and 2.3.4 NEEB calculations section title

Line 298: change "Duncan's test at were" to "Duncan's test were"

Line 306: Add space between Fig 4 caption and paragraph

Line 425: change 202 to proper 4 digit year

Line 602: remove extra Table A1 Header

Line 604: Remove extra Table A2 Header.

Line 605: remove space in inputs

Line 634: Check reference formatting
